# Potential Applications of Thermoresponsive Poly(*N*-Isoproplacrylamide)-Grafted Nylon Membranes: Effect of Grafting Yield and Architecture on Gating Performance

**DOI:** 10.3390/polym15030497

**Published:** 2023-01-18

**Authors:** Todsapol Kajornprai, Putita Katesripongsa, Sang Yong Nam, Zuratul Ain Abdul Hamid, Yupaporn Ruksakulpiwat, Nitinat Suppakarn, Tatiya Trongsatitkul

**Affiliations:** 1Research Center for Biocomposite Materials for Medical Industry and Agricultural and Food Industry, Suranaree University of Technology, Nakhon Ratchasima 30000, Thailand; 2School of Polymer Engineering, Institute of Engineering, Suranaree University of Technology, Nakhon Ratchasima 30000, Thailand; 3Department of Materials Engineering and Convergence Technology, Gyeongsang National University, Jinju 52828, Republic of Korea; 4School of Materials and Mineral Resources Engineering, Universiti Sains Malaysia, Nibong Tebal 14300, Malaysia

**Keywords:** nylon membrane, smart membrane, thermoresponsive polymer, poly(*N*-isopropylacrylamide), water filtration, oil-water separation, gas permeability

## Abstract

This study illustrated the potential applications of thermoresponsive poly(*N*-isopropylacrylamide) (PNIPAm) grafted nylon membranes with different grafting yields and grafting architecture. The thermoresponsive gating performance at temperatures below and above the lower critical solution temperature (LCST) of PNIPAm (32 °C) were demonstrated. The linear PNIPAm-grafted nylon membrane exhibited a sharp response over the temperature range 20–40 °C. The grafting yield of 25.5% and 21.9%, for linear and crosslinked PNIPAm respectively, exhibited highest thermoresponsive gating function for water flux and had a stable and repeatable “open-closed” switching function over 5 cycle operations. An excellent oil/water separation was obtained at T < 32 °C, at which the hydrophilic behavior was observed. The linear PNIPAm-grafted nylon membrane with 35% grafting yield had the highest separation efficiency of 99.7%, while PNIPAm structures were found to be independent of the separation efficiency. In addition, the membranes with thermoresponsive gas permeability were successfully achieved. The O_2_ and CO_2_ transmission rates through the PNIPAm-grafted nylon membranes decreased when the grafting yield increased, showing the better gas barrier property. The permeability ratio of CO_2_ to O_2_ transmission rates of both PNIPAm architectures at 25 °C and 35 °C were around 0.85 for low grafting yields, and approximately 1 for high grafting yields. Ultimately, this study demonstrated the possibility of using these thermoresponsive smart membranes in various applications.

## 1. Introduction

Membrane technology plays an important role in myriad fields, including chemical engineering, biomedical engineering, petrochemical engineering, environmental engineering, food engineering, pharmaceutical engineering, gas separation, water treatment, drug delivery, etc. [1,2,3]. A membrane acts as a selective barrier for regulation/rejection of substances in a mixture. A difference in chemical potential and pressure gradient are the driving forces for allowing specific components through the barrier. A separation process under mild conditions with flexibility in system design and relatively low energy consumption are the main advantages of the membrane [1,2,3]. Although the current membrane technology represents a very significant achievement, the commercially available membranes are still passive. The permeability and permeation performance of the membranes are non-responsive to the changes in environmental conditions. Therefore, the advance toward this application is considered to be restricted [1].

However, stimuli-responsive materials, also known as smart materials, undergo rapid transformation in their microstructure, triggered by changes in their environmental condition. External stimuli can be easily manipulated or controlled, such as temperature, pH, ionic strength, magnetic and electric fields, photo-irradiation, and chemical species [1]. A temperature-responsive membrane is especially interesting because the environmental temperature stimuli can be easily designed and artificially controlled. To fabricate the thermo-sensitive membranes, surface modification by grafting smart polymers onto the membrane’s surface could yield combined advantages of the useful properties of the based membrane and attained functionalities from the grafted polymers. 

Poly(*N*-isopropylacrylamide) (PNIPAm) is the most well-known thermoresponsive polymer which can undergo a discontinuous phase transition and change its configuration upon alteration in temperature. The linear PNIPAm exhibits a thermo-sensitive phase transition at its lower critical solution temperature (LCST) of around 32 °C, which is close to physiological temperature, making it useful for biomedical applications [1] and medical fields such as controlled drug release [4], tissue engineering [5], and immobilization of enzymes [6]. At temperatures below its LCST, the PNIPAm linear chains adopt an extended random coil conformation. The hydrophilic amide (N-H or C=O) groups remain exposed to the water molecules and form intermolecular hydrogen bonds. However, the intramolecular hydrogen bonds disappear when the solution temperature is above its LCST. The formation between amide groups of PNIPAm and water molecules around the polymer chain break and the hydrophobic interactions of the isopropyl groups of PNIPAm govern. As a result, the PNIPAm chains dehydrate and collapse the extended chain [7]. The crosslinked PNIPAm or PNIPAm hydrogels also undergo a volume phase transition temperature (VPTT) at around 32 °C, which is similar to the LCST of linear PNIPAm [8]. Thus, the PNIPAm can change its surface properties from hydrophilicity to hydrophobicity when the temperature increases above its LCST and vice versa. Given the fascinatingly intelligent features of PNIPAm, many researchers designed and fabricated artificial thermoresponsive smart membranes by grafting PNIPAm onto the surfaces of the different porous substrates. The plasma-grafted polymerization method was employed to fabricate the PNIPAm-*g*-HDPE [9], PNIPAm-*g*-PP [10], PNIPAm-*g*-PVDF [11,12], and PNIPAm-*g*-nylon-6 membranes [13], while the acid-assisted and plasma-induced graft polymerization was reported to prepare the PNIPAm-*g*-PC [14].

In our previous work on the synthesis and characterization of PNIPAm-grafted nylon membranes, a new fabrication strategy via a “grafting-from” approach—utilizing the combination techniques of plasma-initiated polymerization and microwave-assisted polymerization—has been reported [15]. The radicals were formed on the surface of porous nylon membrane by argon (Ar) plasma and acted as an initiator for post-graft polymerization. The graft polymerization took place in the solution heated by microwave. Thus, the PNIPAm were chemically attached to the pore surface of the nylon membrane. The microwave output power of 800 W with the irradiation time of 10 min is an optimal microwave condition for the synthesis of PNIPAm-grafted nylon membranes, giving the maximum grafting yield without damaging the nylon membrane substrate [15]. Different grafting architectures, linear and crosslinked PNIPAm, and yields may be useful for different applications. Linear grafted PNIPAm chains have freely mobile ends, resulting in a fast response to temperature stimuli with a sharp transition temperature [16]. On the other hand, the grafted crosslinked structure of the PNIPAm is mechanically stronger than the linear one [17]. It was speculated that the difference in grafting architectures may exhibit different thermoresponsive behaviors and gating characteristics which could be applied in many applications.

The aim of this study is to demonstrate the applications of smart PNIPAm-grafted nylon membranes. The use of hydrophobic materials for oil/water separation and self-cleaning has gained growing attention [18]. PNIPAm-grafted nylon membranes with the “hydrophilic-hydrophobic” switching property are expected to separate the oil/water (O/W) mixtures by alteration of the temperature regulating mechanisms. In addition, the gas transport characteristics—especially oxygen (O_2_) and carbon dioxide (CO_2_)—are very important in the fields of modified atmosphere packaging (MAP). The gas compositions inside the package should be optimal for fresh fruits and vegetables in order to prolong their shelf life and maintain their qualities [19,20]. Therefore, controllable gas permeation with self-regulating gas permeability upon temperature abuse of the PNIPAm-grafted nylon membranes is of interest in this purpose.

In the present study, nylon membranes grafted with two different PNIPAm structures, linear and crosslinked, were synthesized using the method mentioned above [15]. The influences of the grafting yields and PNIPAm architectures on thermoresponsive gating performance were investigated. The thermoresponsive gating properties and the repeatability of grafted nylon membranes were examined by measuring the thermoresponsive water filtration. The utilization of these smart membranes in oil-water separation and control gas permeability were demonstrated.

## 2. Materials and Methods

### 2.1. Materials

The commercial nylon membranes (average pore size of 0.2 μm, 47 mm in diameter, 150–187 mm in thickness) used in this study were purchased from Whatman Co., UK. *N*-isopropylacrylamide (NIPAm; 98% purity) monomer and *N*,*N*′-methylenebisacrylamide (BIS) crosslinker were purchased from TCI Co., Japan. The NIPAm monomer was recrystallized in hexane before being used. Paraffin oil was obtained from ERBApharm, France. Olive oil was purchased from Vidhyasom, Thailand and Tween 20, used as a surfactant, was purchased from OmniPur, USA. Deionized water was used throughout the study.

### 2.2. Synthesis of Linear and Crosslinked PNIPAm-Grafted Nylon Membrane

Plasma-initiated polymerization method coupled with microwave-assisted polymerization technique was employed to graft linear and crosslinked PNIPAm onto the nylon membrane substrates as described previously [15,21]. Briefly, a nylon membrane was treated with Ar plasma at 30 W for 60 s and exposed to oxygen in the atmospheric air for 2 days. In this step, peroxide species were formed on the nylon membrane’s surface. The treated nylon membrane was then brought into contact with a 10 mL NIPAm monomer solution. The monomer solutions used were prepared with concentrations varying from 2–10 wt%. The constant weight ratio of 800:50 (NIPAm:BIS) was added to the solution in the cases of crosslinked structure. The solution, together with the treated membrane, was heated via microwave irradiation to a temperature >60 °C, which was above the decomposition temperature of peroxide groups situated on the treated nylon membrane surface. The alkoxy radicals were generated and acted as an initiator on the nylon membrane surface, initiating the graft polymerization reaction. The microwave output power and irradiation time were kept constant at 800 W and 10 min, respectively. After the irradiation time reached the set point, the PNIPAm-grafted nylon membrane was then washed with deionized water in a temperature-controlled shaking water bath (30 °C) for 12 h to remove any unreacted monomer and homopolymer. The cleaned PNIPAm-grafted nylon membrane was then dried in a hot air oven at 50 °C for 12 h. The grafting yield based on the mass increase ratio after grafting of PNIPAm was calculated according to the following equation: (1)$$Y\left(\%\right)=\frac{W_{g}-W_{0}}{W_{0}}\times 100$$
where *Y* is the grafting yield of PNIPAm on the nylon membrane, and *W_g_* and *W*_0_ are the mass of the nylon membrane after and before grafting, respectively [11]. PNIPAm-grafted nylon membranes with various grafting yields are displayed in Table 1.

### 2.3. Characterizations

#### 2.3.1. Chemical Composition Analysis

To confirm the grafting reaction, functional groups of ungrafted and PNIPAm-grafted nylon membranes were analyzed using an attenuated total reflection Fourier transform infrared spectrophotometer (ATR-FTIR) (Tensor 27, Bruker, Billerica, MA, USA). Spectra were collected in the wavenumber range of 4000 to 400 cm^−1^ with a resolution of 4 cm^−1^ and a number of scans of 64.

#### 2.3.2. Morphological Analysis

The outer surface and cross-sectional microstructures of ungrafted and PNIPAm-grafted nylon membranes were examined using a field emission scanning electron microscope (FE-SEM) (CARL ZEISS-AURIGA, Jena, Germany) at the rotation angle of 32° with a 5 kV accelerating voltage. The membranes were cryo-fractured in liquid nitrogen and sputter-coated with gold for 60 s prior to being examined.

#### 2.3.3. Specific Surface Area Measurement by the Brunauer-Emmett-Teller (BET) Method

BET-specific surface area of ungrafted and PNIPAm-grafted nylon membranes were assessed by a high-performance nitrogen adsorption analyzer (ASAP 2010, Micromeritics, USA). The small pieces of samples at a constant weight of 1 g were used. Degassing pretreatment for BET and pore volume analysis was carried out at 110 °C for 4 h.

#### 2.3.4. Water Contact Angle Measurement

The sessile drop technique was used to determine the water contact angle on nylon and PNIPAm-grafted nylon films. The measurement was performed in an atmospheric air at room temperature using a goniometer (Dino-Lite Digital Microscope, Taiwan). A syringe was used to deposit a 3 µL droplet of deionized (DI) water onto the sample’s surface. The contact angle was determined by analyzing an image captured immediately after the water droplet was placed on the film surface. The reported contact angle of a sample was averaged from five readings. The contact angle of water droplets at temperatures below and above the LCST of PNIPAm were compared. Note that a nylon film was used in place of a nylon membrane to determine the contact angle of the representative without the capillary effect of the pores on the membrane.

#### 2.3.5. Thermoresponsive Water Filtration 

Water permeability measurement of ungrafted and PNIPAm-grafted nylon membranes was carried out at an operating pressure of 100 kPa using filtration apparatus. The direct flow of water feed was carried out using a vacuum pump. The effective area of a membrane for water permeation was 11.34 cm^2^. The temperature of the feedwater was varied from 25 °C to 45 °C. The water flux measuring was used to evaluate the water permeation through different membranes under different temperatures. The water flux (*J*) under the same conditions was obtained an average of three measurements according to equation: (2)$$J=\frac{Q}{A\;\times \Delta T}$$
where Q, A and ∆T were the permeate volume of water (mL), the effective of membrane area (cm^2^), and the permeation time (min), respectively [22].

The “open-closed” switching function, repeatability, and reversibility of the thermoresponsive PNIPAm-grafted nylon membranes were examined. The measurements were performed at temperatures of 40 °C and 25 °C, alternately for 5 cycles. Before each water flux measurement at either 40 °C or 25 °C, the temperature of the feedwater was stabilized at the test temperature for 1 h to ensure the completion of the thermoresponsive phase transition of the grafted PNIPAm gates.

The thermoresponsive gating coefficient (*R*) of the PNIPAm-grafted nylon membrane is the ratio of water flux through the membrane at 40 °C relative to that at 25 °C. The thermoresponsive gating coefficient was determined using the following equation: (3)$$R=\frac{J_{40}}{J_{25}}$$
where *J*_40_ and *J*_25_ are the water fluxes at feedwater at temperatures of 40 °C and 25 °C, respectively. When the water fluxes at 25 °C and 40 °C are identical, a gating coefficient of 1 indicates that a membrane has no thermoresponsive gating function [13].

#### 2.3.6. Temperature-Responsive Controllable Oil/Water Separation

The ability of PNIPAm-grafted nylon membrane for oil/water separation depended on the grafting yields, the PNIPAm structures, and the environmental temperatures. The smart membrane could separate oil-in-water (O/W) and water-in-oil (W/O) emulsions at temperatures below and above the LCST of PNIPAm, respectively. 

O/W emulsions with two different oils (paraffin and olive oil) were prepared with a ratio of 5:95 *w*/*w*. Tween 20 surfactant 1 mg/mL was added to stabilize the emulsions. The mixture was strongly stirred using a magnetic stirrer before being shaken until a milky solution was obtained (about 48 h). An oil-water mixture with a ratio of 60:40 *w*/*w* was prepared by a similar method.

The PNIPAm-grafted nylon membrane was installed as a filter in the 41 mm inner diameter laboratory filtering apparatus. The 5 wt% O/W emulsion was poured into a glass funnel at 25 °C, and separation was accomplished simply by gravity. For T > LCST, the swollen membrane was submerged in the 45 °C water to deswell. Any excess water was removed from the membrane’s surface using tissue paper. To perform the separation, the 60 wt% oil-water mixture was heated to 45 °C before being poured into the separation apparatus, which was placed in a 45 °C oven. After separation, the amount of remaining oil in the collected water was determined by evaporating the water in an oven and weighing the leftover oil. The following equation was used to calculate the separation efficiency: (4)$$S\left(\%\right)=\frac{W_{0}-W_{1}}{W_{0}}\times 100$$
where *S* is separation efficiency, *W*_0_ (g) is the initial oil or water weight in the feed, and *W*_1_ (g) is the oil or water weight in permeate after filtration [23].

#### 2.3.7. Thermoresponsive Gas Permeability Control 

The effects of temperature variation on gas permeability of PNIPAm-grafted nylon membrane were investigated by the static method, similar to those previously described in the literature [24,25]. The oxygen and carbon dioxide transmission rates (O_2_TR and CO_2_TR) of PNIPAm-grafted nylon membrane were measured below and above the LCST of PNIPAm. The membrane was immersed in 25 °C DI water for 20 min to allow swelling of PNIPAm chains. The membrane was placed on the top lid of a gas-impermeable acrylic cylindrical container (1,179 cm^3^ capacity) having a 10 mm perforated diameter size, as shown in Figure 1. The gas inside the container was evacuated for 3 min by a high vacuum pump through a needle that was inserted into the rubber septum, and then the 21% CO_2_ gas mix (balanced N_2_) was introduced for 5 min. After finishing two-cycle replications, the needle was removed, and the container was placed in the 25 °C air-conditioned room. For the collapse state of PNIPAm, the procedure was done similarly to that of the swollen state of PNIPAm, with the exception that the membrane was immersed in 35 °C DI water and the container was placed in the oven at 35 °C without air circulation. To measure the gas permeability, gases inside the container of 5 cm^3^ were taken by a syringe every hour and 5 cm^3^ N_2_ was simultaneously added during sampling. This was to keep a constant volume of gas in the container, and, thus, prevent the pressure driven permeability due to the difference of pressure between the outside and inside of the chamber. After that, the gas compositions were analyzed by a gas chromatograph (7890A, Agilent Technologies, Santa Clara, CA, USA). Gas transmission rate per volume of the cylinder can be obtained from the slope of the plot of $\ln\frac{\left(C_{i,in}\left(t\right)-C_{i,out}\right)}{\left(C_{i,in}\left(0\right)-C_{i,out}\right)}$ versus time, where $C_{i,in}$ and $C_{i,out}$ are the volumetric fraction of gas i inside and outside the container at the initial time (0) and at a determined time (t), respectively [24,25].

## 3. Results and Discussion

### 3.1. Surface Functionalization of Nylon Membranes

#### 3.1.1. Compositional Verification of PNIPAm-Grafted Nylon Membranes

The intention of a surface modification of a nylon membrane is to introduce new functionalities to the membrane in order to improve its performance. Figure 2 shows the ATR-FTIR spectra of ungrafted and PNIPAm-grafted nylon membranes. The grafting yields calculated from Equation (1) were indicated on each spectrum. Both ungrafted and PNIPAm-grafted nylon membranes exhibited characteristic peaks of the acrylamide functional group (HN-C=O) at 1540 (N-H bending), 1650 (C=O stretching), and 3350 (N-H stretching) cm^−1^ in their FTIR spectra. Therefore, these peaks were unsuitable to be used to differentiate PNIPAm from nylon. Consequently, the graft polymerization reaction was confirmed by following the methyl group of PNIPAm’s isopropyl group. FTIR spectra of both linear and crosslinked PNIPAm-grafted nylon membranes revealed a new absorption band at 2970 cm^−1^ (C-H stretching, methyl group), along with double peaks at 1385 cm^−1^ and 1368 cm^−1^ (symmetrical bending vibrations and coupling split originating from the bimethyl group of the isopropyl group of PNIPAm) [17,26,27]. In addition, the intensity of these three notable peaks increased when the grafting yield increased. These results supported the notion that PNIPAm were successfully grafted on the nylon membranes by the combined techniques of plasma-initiated polymerization and microwave-assisted polymerization.

#### 3.1.2. Morphological Analysis of PNIPAm-Grafted Nylon Membranes

To understand the effect of surface modification on the thermoresponsive performance of the PNIPAm-grafted nylon membrane, surface morphologies of ungrafted and PNIPAm-grafted nylon membranes with different grafting yields and grafting architectures were observed. FE-SEM micrographs of cross-sectioned surfaces of the membranes are displayed in Figure 3. The ungrafted nylon membrane in Figure 3a has a honeycomb-like porous structure beneath the skin [13]. The morphology was notably different from those of PNIPAm-grafted nylon membranes. Both linear and crosslinked PNIPAm were grafted locally to the membrane surface and pore entrance, as shown in Figure 3b,c.

As grafting yield increased (Figure 3d,e), the pore opening channels on the membrane’s top surface became smaller and it seemed to be denser. This indicated that PNIPAm molecules were grafted not only on the surface, but also on the shallow interior of the pores over the entire membrane thickness. Similar findings have been reported in the literature that the grafted PNIPAm was predominantly on the surface of the membrane substrate and appeared to fill the pores close to the membrane surface [11,13,17]. This could be because the hydrophobic PNIPAm oligomer was in a shrunken state at synthesis temperatures above its LCST, allowing the PNIPAm chains to accumulate and obstruct the monomer migration to the pore during polymerization [28]. Due to the influence of the propagating chains, the grafted PNIPAm layer on the inner pore surface was of insignificant thickness. Other observations also demonstrated that the grafting of PNIPAm did not alter the substrate’s porosity nature, but merely coated the inside and outside pores [29]. However, there were contradictory findings in the published research of Choi, Y. J. et al., who reported that the grafting of PNIPAm enlarged the pore size of the polypropylene (PP) membrane [30].

#### 3.1.3. Surface Characteristics of PNIPAm-Grafted Nylon Membranes

This part addressed the change of surface characteristics of the nylon membranes after grafting with PNIPAm. The BET-specific surface area and total pore volume of linear PNIPAm-grafted nylon membranes with varying grafting yields are shown in Table 2. The specific surface areas and pore volumes of the membranes dropped marginally as grafting yield increased. These indicated an increase in the number of PNIPAm grafts. Choi, Y. J. et al. similarly observed that the specific surface area and pore volume of the PNIPAm-grafted PP membrane decreased as a function of an increasing grafting quantity [30]. 

The “hydrophilic-hydrophobic” switching property of PNIPAm-grafted nylon membrane has become a hot issue due to its thermally responsive conformation changes with self-adjustment of the PNIPAm segments. The grafting of PNIPAm to the surface of a nylon membrane altered not only the membrane’s surface morphology and pore sizes, but also its wettability. Nonetheless, the honeycomb structure of the nylon membrane prevented it from meeting the requirements for measuring the water contact angle; the capillary force caused the water droplets to migrate toward the pore connecting channels. Therefore, nylon film was used to substitute nylon membrane for water contact measurement. The PNIPAm-grafted nylon films were produced using the same procedure as the PNIPAm-grafted nylon membranes. Figure 4 presents the water contact angles of ungrafted, linear, and crosslinked PNIPAm-grafted nylon films at temperatures below and above the LCST of PNIPAm, and the values of water contact angle of PNIPAm-grafted nylon films with various grafting yields were summarized in Figure 5. In the case of PNIPAm-grafted nylon film of both architectures with low grafting yields, the contact angle values were similar. Thus, the one-way ANOVA and Tukey’s test (*p* < 0.05) were used to investigate the effect of different grafting yields on the water contact angle. The water contact angle of the ungrafted nylon film was 85.6 ± 2°, independent of the measuring temperature. One-way ANOVA exhibited a statistically significant difference among the means of water contact angle at the temperature of 25 °C of both linear and crosslinked PNIPAm-grafted nylon films (*p*-value was <0.00001). The multiple pairwise comparisons with Tukey’s test revealed there was no significant difference in mean of water contact angle between the ungrafted nylon film and 1% grafting yield of linear PNIPAm (*p* = 0.11) as well as the crosslinked samples with the grafting yield of 1 and 6% (*p* = 0.58). This means that the low grafting yields did not enhance the hydrophilicity of the nylon films. With regard to increasing the temperature, all PNIPAm-grafted nylon films had lower water contact angles at 25 °C than those at 35 °C. This indicated that the hydrophilicity of PNIPAm-grafted nylon film decreased, as the temperature was increased above its LCST. Additionally, the difference between the water contact angles of PNIPAm-grafted nylon films at different testing temperatures became more pronounced when the grafting yield increased. This was a result of the increased coverage of the grafted PNIPAm. The Tukey’s test revealed that the water contact angle of all crosslinked PNIPAm-grafted nylon films at 35 °C showed no significant differences between pairwise comparisons (*p* > 0.05). The presence of a crosslinker restrained the chain mobility of the PNIPAm, and the intermolecular interaction between PNIPAm chains and water molecules was difficult to get ascendancy over the polar groups [31,32]. As a result, the change in contact angle was less prominent. These results demonstrated that the linear PNIPAm structure had a profound effect on improving the hydrophilicity of the film surface more than the crosslinked PNIPAm.

Figure 6 shows the water contact angle measurement of highly PNIPAm-grafted nylon membranes (grafting yield > 100%). The pulling of the water droplets caused by capillary force was not observed which was possible due to the blockages of the membrane pores by the grafted PNIPAm. Thus, the PNIPAm-grafted nylon membranes were used in this case and the water contact angles of the PNIPAm-grafted nylon membranes were compared with the ungrafted nylon film (assume as membrane). The results showed that the highly PNIPAm-grafted nylon membrane of both architectures had a very small change in response to the temperature, as compared with the ungrafted nylon film. This could be attributed to the chain mobility restriction of the hydrophilicity from the curling of the linear PNIPAm chains and the chain conformation hindering the crosslinked network. As a result, the switchable surface wettability at the temperature below and above its LCST of the highly PNIPAm-grafted nylon membranes had lesser effectiveness than those of the lower grafting yields.

### 3.2. Thermoresponsive Water Filtration Application

To demonstrate the potential applications of the PNIPAm-grafted nylon membranes on thermoresponsive water filtration, the water flux measurement was employed to evaluate the membrane permeability. Figure 7 depicts the effect of temperature on the water permeability of the linear and crosslinked PNIPAm-grafted nylon membrane with different grafting yields, ranging from 20 °C to 45 °C. The results showed that the PNIPAm-grafted nylon membranes of both grafting architectures showed lower water flux compared to the ungrafted nylon membranes, as presented in Figure 6a,c. This is due to the additional thickness imparted by the PNIPAm layers. When the temperature increased from 20 °C to 45 °C, the flux values of PNIPAm-grafted nylon membranes of linear and crosslinked structures tended to increase. These phenomena, caused by the phase transition of PNIPAm, lead to the alteration of the water permeability of the modified membranes. In general, water flux is independent of temperature. However, the experimental results showed that the water flux of the ungrafted nylon membrane (0% grafting yield) increased with an increase in the feedwater temperatures from 25 °C to 45 °C. The aforementioned result was due to the reduction in liquid viscosity coefficient with increasing temperatures [13]. To eliminate the effect of the change in liquid viscosity coefficient as the temperature increased and focus mainly on thermoresponsive properties of the PNIPAm-grafted nylon membranes, water fluxes were normalized versus the reference temperature (20 °C), using the following equation: (5)$$J_{T_{0}}=J{\left[\frac{42.5+T_{0}}{42.5+T}\right]}^{1.5}$$
where *J* is flux observed, $J_{T_{0}}$ is flux at a reference temperature, *T* is water temperature, and *T*_0_ is reference temperature (20 °C) [33].

Figure 7b,d presents the normalized water flux of PNIPAm-grafted nylon membranes and it was observed that the water fluxes through ungrafted nylon membrane showed rather similar values over the entire temperature range. The normalized water fluxes of the linear PNIPAm-grafted nylon membranes with the low and medium grafting yields of 9, 10.7 and 22.5% suddenly increased in the temperature range from 30 °C to 35 °C, corresponding to the LCST range of PNIPAm. PNIPAm-grafted polycarbonate track-etched (PCTE) membranes have been found to behave in a similar manner. The water flux of the PNIPAm-grafted PCTE membranes in the temperature range of 25 °C to 30 °C was much lower than that in the temperature range of 34 to 40 °C. A sharp transition in the water flux occurred as the temperature increased from 30 °C to 34 °C [34]. This sharp increase behavior could only be due to the pore opening becoming suddenly enlarged. This was thought to be due to the shrinking of the grafted PNIPAm chains on the pore surfaces, as they were in the hydrophobic state when the feedwater temperature was above the LCST. As a result, the pore size increased—in other words, the pores were opened by the PNIPAm gates which allowed water to permeate through with higher water flux. In contrast, when the temperature of the feedwater was below the LCST, the grafted PNIPAm chains on the inner pore surfaces were in the swollen state and closed off the gates. From this result, it suggested that the linear PNIPAm chains grafted in the membrane pores possessed the ability to act as smart thermoresponsive gates. However, it was also observed that the thermoresponsive characteristics of the water flux of the linear PNIPAm-grafted nylon membranes with different grafting yields were different. In the case of PNIPAm-grafted nylon membrane with high grafting yield, such as at 43.2%, the thermoresponsive property was diminished. Similar results were also found in the PNIPAm-grafted PCTE membranes in which the water flux became zero, showing no thermoresponsive gating characteristics as the pore-filling ratio was larger than 44.2% [34]. This might be due to the pores of the membrane being overly filled. Too many linear PNIPAm chains were grafted inside the pores. The polymer chains might be tightly packed inside the pore, disabling the chain movement during the coil-globule conformation change. As a result, the membranes with highly grafted yield were unable to function as a thermoresponsive membrane even when stimulated with a temperature above LCST.

Figure 7d shows the normalized water flux over the temperature rate of 20–45 °C of the crosslinked PNIPAm-grafted nylon membranes. It can be clearly seen that the thermoresponsive properties of these membranes were relatively weak, as compared to those of linear grafted nylon membranes (Figure 7b). This was because the crosslinked network structures of grafted PNIPAm layers were more compact than those of the linear PNIPAm chains with the presence of free ends. At the same grafting yield, such as over the range of 9–20%, the effective pore size of the crosslinked PNIPAm-grafted nylon membranes at 25 °C was larger than that of linear PNIPAm-grafted nylon membranes. Therefore, the water flux of the crosslinked PNIPAm-grafted nylon membranes at this temperature (below LCST) was higher than that of the linear ones. However, a sharp response in the water flux of these crosslinked PNIPAm-grafted nylon membranes was not observed over the entire temperature range. This was due to the crosslinked structure preventing the PNIPAm networks from fully swelling, even at temperatures below the LCST of PNIPAm [17,35].

Figure 8 shows the thermoresponsive gating coefficient (*R*) of the linear and crosslinked PNIPAm-grafted nylon membrane as a function of grafting yield. The gating coefficient value of the ungrafted nylon membrane was assumed to be 1.0 because the ungrafted nylon membrane was considered to be a non-thermoresponsive membrane. Therefore, a higher value of the gating coefficient should indicate the better thermoresponsive properties of the membranes. For the linear PNIPAm-grafted nylon membranes, the gating coefficient increased as the grafting yield increased up to 22.5%. The maximum gating coefficient was estimated from the trend line and was approximately 8. As the grafting yield further increased, the gating coefficient value decreased and become 1 (non-thermoresponsive) at grafting yield greater than 40%, as the pores on the membrane were choked by the grafted PNIPAm molecules. On the other hand, the relationship between gating coefficient and grafting yield was quite different for the crosslinked PNIPAm-grafted nylon membranes. The critical grafting yield of PNIPAm for choking the membrane pores was as high as 92.7%, while the optimum grafting yield for thermoresponsive gating was 21.9%. However, it should be noted that the largest gating coefficient of the crosslinked PNIPAm-grafted nylon membrane was a mere 1.37. The inserted picture in Figure 8 shows the comparison of the thermoresponsive gating coefficient between linear and crosslinked PNIPAm-grafted nylon membranes. It was clear that thermoresponsive “open-closed” switching performance for the linear PNIPAm-grafted nylon membrane is considerably better than that for the crosslinked PNIPAm-grafted nylon membrane at the optimal grafting yield.

From the abovementioned results, many factors influenced the thermoresponsive gating characteristics of the PNIPAm-grafted nylon membrane, including grafting yields and grafting architectures. However, the hydrophobic/hydrophilic properties and the porous microstructure of the membrane substrates were also discernible to the thermoresponsive water permeability. Yang et al. compared the effect of grafting PNIPAm on the hydrophilic nylon membrane and poly(vinylidene fluoride) (PVDF) porous membranes on the thermoresponsive gating characteristics [13]. The hydrophobic porous PVDF membrane with closely packed finger-like large cavities had a negative effect not only on the water flux, but also on the thermoresponsive gating coefficient. At the same grafting yields and feedwater temperature, the water flux and gating coefficient value of the PNIPAm-grafted nylon membrane was always much higher than that of the PNIPAm-grafted PVDF membrane. Nylon membrane features a porous honeycomb structure below the skin layer. The nylon membrane is hydrophilic in nature which can be effectively grafted with PNIPAm and exhibited a highly thermoresponsive gating performance.

In certain applications, a membrane may be used repeatably. The stability of the PNIPAm-grafted nylon membranes in long-term operations is preferred. To investigate this attribute of our samples, the reversibility of the thermoresponsive “open-closed” switching function of the membranes was examined. Figure 9 shows the switchable gating function characteristic of PNIPAm-grafted nylon membranes with the grafting yield of 22.5% for linear and 21.9% for crosslinked PNIPAm structures, selecting from the largest thermoresponsive gating coefficient value. It can be seen that both linear and crosslinked PNIPAm-grafted nylon membranes experienced stable gating characteristics within 5 cycles of operation and underwent repeated temperature switching between 40 °C and 25 °C. The water fluxes of both membranes were nearly constant, even after 5 consecutive switching cycles. A good, repeatable water permeation performance was observed for the polyether sulfone composite membranes blended with PNIPAm nanogel, showing the great reversible thermoresponsive water flux with 3 cycles [36]. PNIPAm-grafted PET membrane also showed the good thermoresponsive repeatability with almost-constant water flux during 5 cycle replications [37]. In addition, a similar finding of excellent stability thermoresponsive permeation after 5 repeating cycles was also found in the case of PVDF ultrafiltration (UF) membrane incorporated with PNIPAm-grafted silica nanoparticles (SiO_2_-PNIPAm) [38]. This result suggested that the thermoresponsive “open-closed” switching was satisfactorily reversible and stability of these thermoresponsive gating membranes, as well as water permeation repeatability of PNIPAm-grafted nylon membrane, was effectively achieved.

### 3.3. Oil-Water Separation Application

With environmental pollution, oil contaminated water in the forms of oil-water mixtures or emulsions have become a worldwide problem. Many separation processions are extensively developed. Therefore, the oil-water separation, acquired from the thermoresponsive “hydrophilic-hydrophobic” switching property of the PNIPAm-grafted nylon membrane, was demonstrated in this study. A hydrophilic membrane was suitable for O/W emulsion separation while a hydrophobic membrane was appropriate for water-in-oil emulsion separation. For O/W emulsion separation, the temperature below the LCST of PNIPAm was performed. Figure 10a shows the separation efficiency of paraffin oil and olive oil in water through the linear PNIPAm-grafted nylon membrane with different grafting yields at 25 °C. It can be seen that the 5 wt% paraffin and olive oils in water emulsion with the addition of Tween 20 surfactant were successfully separated with high separation efficiency of over 80%. The hydrophobic nature of oil has an effect on the separation ability of the PNIPAm-grafted nylon membranes. From the experimental results, the contact angle of olive oil (19.14 ± 3°) on the nylon film at 25 °C was significantly lower than that of the paraffin oil (28.55 ± 4°), implying that olive oil was more hydrophobic than paraffin oil. Thus, the intermolecular repulsion of the hydrophilic PNIPAm-grafted nylon membrane and olive oil was stronger than that of paraffin oil. As a result, the olive-oil-in-water emulsion showed a higher separation efficiency than that of the paraffin oil. Furthermore, the excellent separation efficiency of 99.7% was found when the grafting yield increased up to 35%, caused by the decrease in pore size. Therefore, only water was selectively permeated through the PNIPAm-grafted nylon membrane.

Figure 10b shows the effect of PNIPAm architectures on the O/W separation efficiency of the PNIPAm-grafted nylon membranes with the same grafting yield (~10%). The finding revealed that the separation efficiency of two oils for the ungrafted nylon membrane was close to that of both linear and crosslinked PNIPAm-grafted nylon membrane. This was attributed to the nylon membrane exhibiting hydrophilicity in nature. As a result, oils could not penetrate through the nylon membrane. In addition, it can be seen that the separation efficiency of linear and crosslinked PNIPAm-grafted nylon membrane was not significantly different. Therefore, the aforementioned results inferred that the grafting yields played a key role in the O/W separation and the PNIPAm structures were independent of separation efficiency.

Another interesting discussion is on the separation of the W/O emulsion system at the temperature above the LCST of PNIPAm. However, the W/O emulsion separation experiment was not conducted in this study. Research on the PNIPAm-grafted porous membrane exists in the literature. In the work of Ou et al. [23], polyurethane (TPU) microfiber web with the loaded 3.6 wt% PNIPAm hydrogel showed an excellent ability to separate 99 wt% silicone oil-water emulsions at 45 °C with a separation efficiency of ≥99.26%. It is worth emphasizing that PNIPAm-grafted nylon membrane could possibly be used, for example, in an application of hot water-oil emulsion separation from the crude distillation industry without the necessary extra thermal energy being added.

Apart from O/W emulsion separation, a 60 wt% oil-water mixture—which was not emulsion—was also demonstrated. To clarify the separation efficiency of the PNIPAm-grafted nylon membrane at the hydrophobic state, the paraffin oil and linear PNIPAm structure were used in this experiment because the hydrophilicity of paraffin oil was closer to the hydrophilicity of water than that of the olive oil and the linear PNIPAm had a sharper transition from swollen to shrunken state around its LCST than that of the crosslinked one, respectively. Figure 11 shows the separation efficiency of the water-in-oil through the linear PNIPAm-grafted nylon membrane with different grafting yields at 45 °C. It was interesting to find that both oil and water did not permeate through the PNIPAm-grafted nylon membrane. Comparing the oil-water mixture separation to that of the water flux results (Figure 7a,b), at the particular grafting yield of 10 and 35% of linear PNIPAm, it should be noted here there was a difference in the test methodology. The water flux was done under pressure while the water in the oil-water mixture permeated gravimetrically. Hence, the water could pass through the PNIPAm-grafted nylon membrane in the water flux experiment which was retained in the case of oil-water mixture separation. The possible explanation for the retained oil-water mixture above the PNIPAm-grafted nylon membrane was that the PNIPAm chains on the surface of the nylon membrane restructured themself when the temperature was above its LCST, resulting in hydrophobic properties. The oil—having density lower than that of water—floated on the top of the water surface. As a result, the separation efficiency of less than 1% was obtained. This phenomenon was profoundly affected when the grafting yield increased.

Figure 12 demonstrates the gravimetrical separation of 50 mL paraffin oil (upper phase) and 100 mL water (lower phase) mixture through the linear PNIPAm-grafted nylon membrane with the grafting yield of 35%. The water was mixed with red food coloring and the experiment was started from the hydrophobic state of the PNIPAm-grafted nylon membrane at the water temperature of 45 °C. The results showed that both water and oil phases remained above the membrane after being placed in a 45 °C oven for 2 h. The oil top layer can be separated by cooling down the environment temperature to below the LCST of PNIPAm. The water completely permeated through the PNIPAm-grafted nylon membrane within 24 h, while the oil remained above the membrane. Thus, the oil and water phases in the mixture were successfully separated. These findings are very beneficial for the oil strainer application with a thermally controllable separation.

### 3.4. Self-Regulating Gas Permeability Application 

To the best of our knowledge, there is currently no information available on the gas permeation of the PNIPAm-grafted nylon membrane. The gas transport characteristics, especially oxygen (O_2_) and carbon dioxide (CO_2_), are very important in the fields of modified atmosphere packaging (MAP). The gas compositions inside the package should be optimal for each fresh fruit and vegetable, to prolong their shelf life. During storage and transportation, temperature fluctuation can happen, resulting in a change in the respiration rate of the fresh produce and loss of quality [19,20]. A packaging with self-regulating gas permeability upon temperature abuse is favorable. The phase transition temperature of PNIPAm at around 32 °C could be used to as a mechanism to control gas permeability of a packaging in poor temperature-controlled logistics. 

To expand feasibility studies, the gas permeability of the PNIPAm-grafted nylon membrane was evaluated. This part addresses the effects of environmental temperature, grafting yield, and PNIPAm architectures on the O_2_ and CO_2_ transmission properties. The O_2_TR, CO_2_TR, and the permselectivity or permeability ratio of CO_2_TR to O_2_TR (denoted as β) are summarized in Table 3. The experimental data showed that the ungrafted nylon membrane had O_2_TR and CO_2_TR at 25 °C of about 228 ± 26 and 192 ± 14 cm^3^/h, respectively. Many researchers generally accept that temperature has an influence on gas permeability, especially for non-porous film. The gas transmission rate of the non-porous film linearly increases with increasing temperature, following the Arrhenius equation [39,40,41,42]. However, the influence of storage temperature on the gas transmission rate of perforated film or porous film is quite different. The permeation of gases mainly takes place through the perforations or the connecting channels, resulting in reduced effect and no definite trend on gas transmission rates [42,43,44]. Therefore, the O_2_TR and CO_2_TR at 35 °C of ungrafted nylon membrane gases slightly increased when the temperature increased to 35 °C.

Analysis of the PNIPAm-grafted nylon membranes showed that the permeability temperature relationship depended on the grafting yield and PNIPAm architectures. At a low grafting yield (~11%), the O_2_TR and CO_2_TR of nylon membrane with crosslinked PNIPAm structure at 25 °C were lower than that of the linear one. This was related to the microstructure of the PNIPAm-grafted nylon membrane. As described previously, the crosslinked PNIPAm had a compact with the continuous network coated on the surface of the membrane, while linear PNIPAm displayed as discontinuous islands with irregular shape. Thus, the gas transportation through the connecting holes of crosslinked PNIPAm-grafted nylon membrane was more difficult when compared to the linear grafted one. In addition, it could be seen that with an increase in grafting yield of both linear and crosslinked PNIPAm structures, the effective permeability at 25 °C decreased. A possible explanation for this phenomenon was that the PNIPAm chains were overfilled and stuffed in the pore connecting channels. As a result, the diffusions of O_2_ and CO_2_ through the PNIPAm-grafted nylon membrane were relatively difficult and the O_2_ and CO_2_ permeabilities of the membrane were rather low.

The temperature also affected the gas transmission rate of the PNIPAm-grafted nylon membrane. In the case of low grafting yield (~11%) the O_2_TR and CO_2_TR of both linear and crosslinked PNIPAm structures significantly increased by increasing the temperature from 25 to 35 °C, due to the collapsing of PNIPAm chains above its LCST. An interval of O_2_TR and CO_2_TR between 25 and 35 °C of linear PNIPAm-grafted nylon membranes was larger than the values of the crosslinked PNIPAm structure. This was because the crosslinked PNIPAm networks limited the chain mobility during swell and collapse processes with respect to temperature. Thus, the pore opening sizes allowing the gas transportation of the crosslinked PNIPAm-grafted nylon membrane were minimized. As a result, the gas transmission rates deteriorated. In the case of high grafting yield, both linear and crosslinked PNIPAm-grafted nylon membranes showed that the O_2_TR and CO_2_TR readings at both 25 °C and 35 °C of the linear were indifferent. The readings were lower than those of the membranes with low grafting yield. This result showed the better gas barrier properties of these membranes. In addition, there were no significant positive effects with temperature. The interval of O_2_TR and CO_2_TR at 25 and 35 °C of both PNIPAm structures was narrow due to the disabled chain movement of the dense PNIPAm chains constrained in open connecting pores.

Under nearly all conditions, there was no significant difference in the permselectivity of the PNIPAm-grafted nylon membranes. The results for the linear and crosslinked PNIPAm-grafted nylon membranes with low grafting yield were approximately 0.85, which was in the same range of the single perforated film (0.8–0.9) reported in the literature [42,43,45,46]. However, the permselectivity of linear and crosslinked PNIPAm-grafted nylon membranes with a high grafting yield was close to 1. The results indicated that it was possible to use PNIPAm-grafted nylon membranes with different grafting yield to change the gas composition in a packaging.

## 4. Conclusions

This study demonstrated the potential applications of PNIPAm-grafted nylon smart membranes. The effect of grafting yields and PNIPAm architectures of the smart membranes were found to be crucially important in influencing their performance in thermoresponsive “open-closed” switching functions, as well as thermoresponsive “hydrophilic-hydrophobic” switching properties. PNIPAm-grafted nylon membranes exhibited distinct wettability at different temperatures, below and above the LCST of PNIPAm. The linear PNIPAm chains with the free ends could move relatively more freely as compared to the crosslinked ones, resulting in a lower water contact angle. Linear and crosslinked PNIPAm-grafted nylon membranes with high grafting yield over 100% had no significant change in water contact angle. The thermoresponsive water filtration had no function when the grafting yields were larger than 43.2% for a linear PNIPAm and 92.7% for a crosslinked structure. The thermoresponsive gating coefficient of PNIPAm-grafted nylon membranes with the grafting yield of 25.5% for linear and 21.9% for crosslinked PNIPAm structures were the critical grafting yields for the water flux, showing the highest thermo-responsive gating coefficient (*R*). The membranes with grafting yield greater than these critical values became non-response to temperature change because the blocked pathways prevented water from passing through the membranes at any temperature. Both linear and crosslinked PNIPAm gates in the grafted nylon membranes were proved to be robust and stable as they could repeatedly perform thermoresponsive “open-closed” switching function with maintained water flux at temperatures either below and above the LCST of PNIPAm over 5 cycles, under operation pressure of 100 kPa. Moreover, a linear PNIPAm-grafted nylon membrane with the grafting yield of 35% showed an excellent separation efficiency of 99.7% for the separation of olive oil in water. In addition, the feasibility studies on gas permeability through the PNIPAm-grafted nylon membrane revealed that the magnitude of O_2_TR and CO_2_TR increased with an increase in temperature. The O_2_TR and CO_2_TR interval between two temperatures below and above the LCST of PNIPAm could be altered by varying the grafting yields as well as PNIPAm structures. It is noteworthy that the PNIPAm-grafted nylon membranes have the potential to be used in many applications as an alternative material.

## Figures and Tables

**Figure 1 polymers-15-00497-f001:**
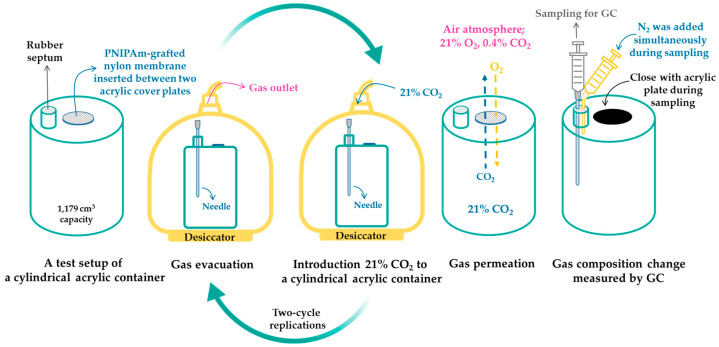
Schematic of gas-impermeable acrylic cylindrical container and gas permeation measurement.

**Figure 2 polymers-15-00497-f002:**
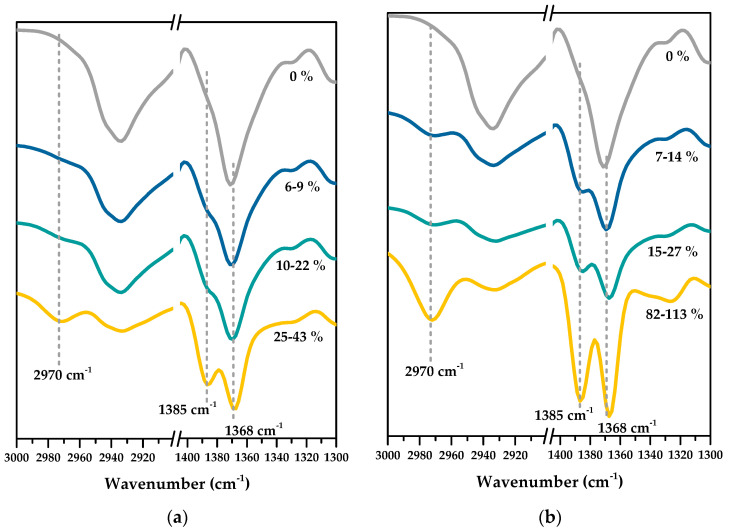
ATR-FTIR spectra of PNIPAm-grafted nylon membranes with different grafted yields for: (**a**) linear; (**b**) crosslinked structures.

**Figure 3 polymers-15-00497-f003:**
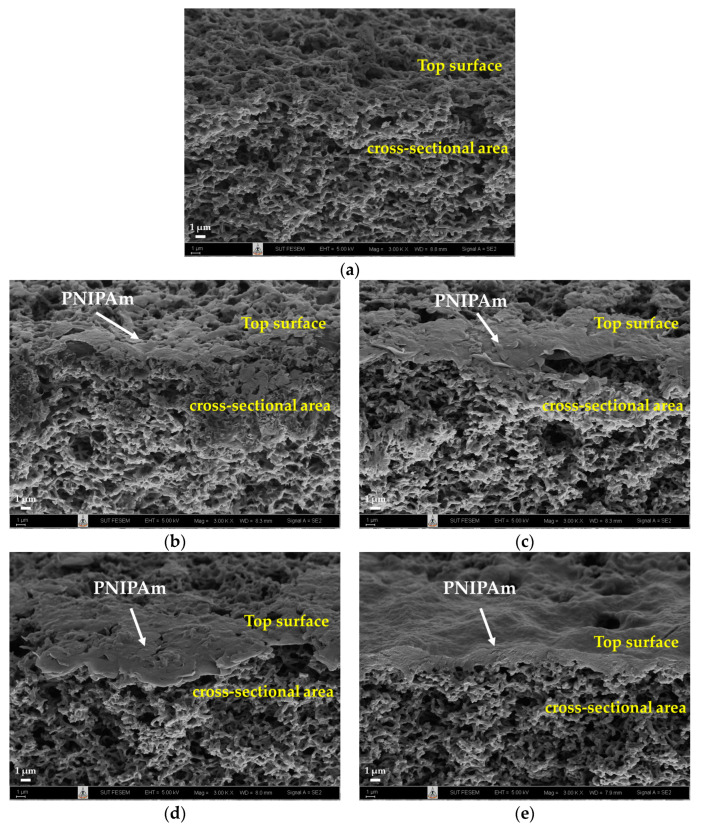
FE-SEM micrographs of cross-sections of ungrafted, linear, and crosslinked PNIPAm-grafted nylon membranes: (**a**) ungrafted nylon membrane; (**b**) linear PNIPAm-grafted nylon membranes (*Y* = 22.5%); (**c**) crosslinked PNIPAm-grafted nylon membranes (*Y* = 21.9%); (**d**) linear PNIPAm-grafted nylon membranes (*Y* = 43.2%); (**e**) crosslinked PNIPAm-grafted nylon membranes (*Y* = 92.7%).

**Figure 4 polymers-15-00497-f004:**
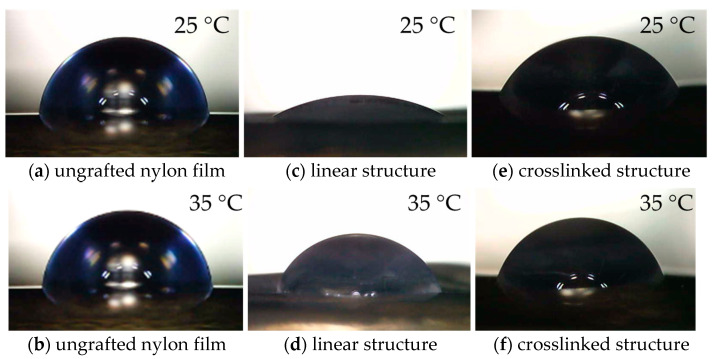
Water contact angles images at 25 and 35 °C of (**a**,**b**) ungrafted nylon film, (**c**,**d**) linear structure, PNIPAm-grafted nylon film (*Y* = 16%) and (**e**,**f**) crosslinked PNIPAm-grafted nylon film (*Y* = 10%).

**Figure 5 polymers-15-00497-f005:**
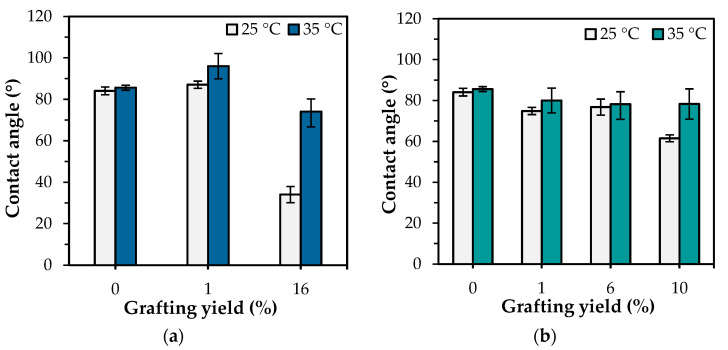
Water contact angles of ungrafted and PNIPAm-grafted nylon film with different grafting architectures ass a function of grafting yield at 25 and 35 °C: (**a**) linear; (**b**) crosslinked structures.

**Figure 6 polymers-15-00497-f006:**
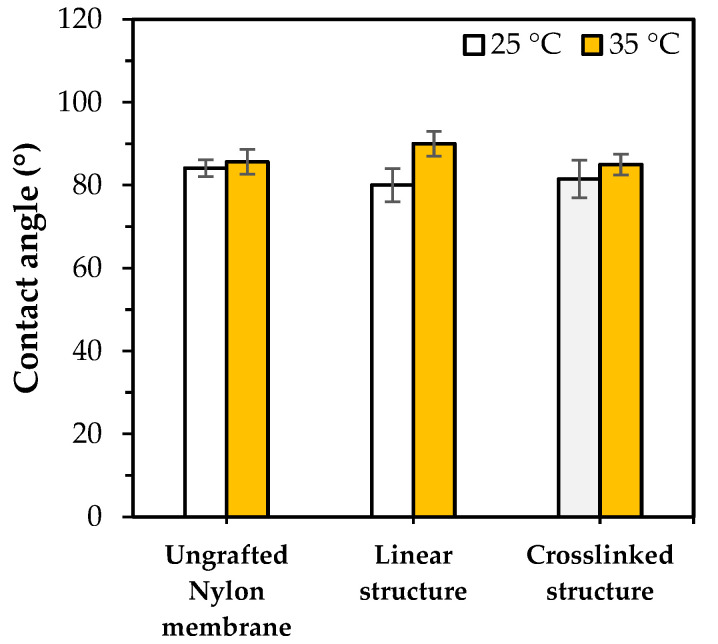
Water contact angles of ungrafted and highly PNIPAm-grafted nylon membranes (*Y* > 100%) with different grafting architectures at 25 °C and 35 °C.

**Figure 7 polymers-15-00497-f007:**
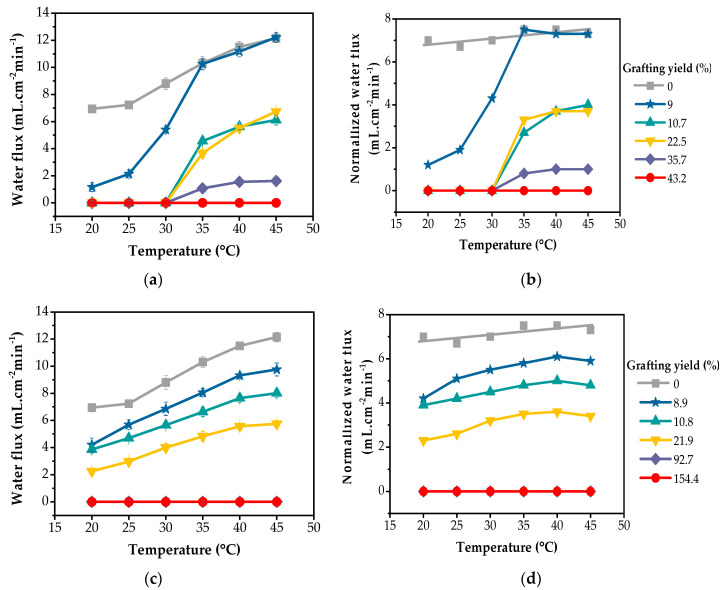
Thermoresponsive characteristics of water flux through PNIPAm-grafted nylon membranes with different grafting yields before and after normalization of: (**a**,**b**) linear; (**c**,**d**) crosslinked structure, respectively.

**Figure 8 polymers-15-00497-f008:**
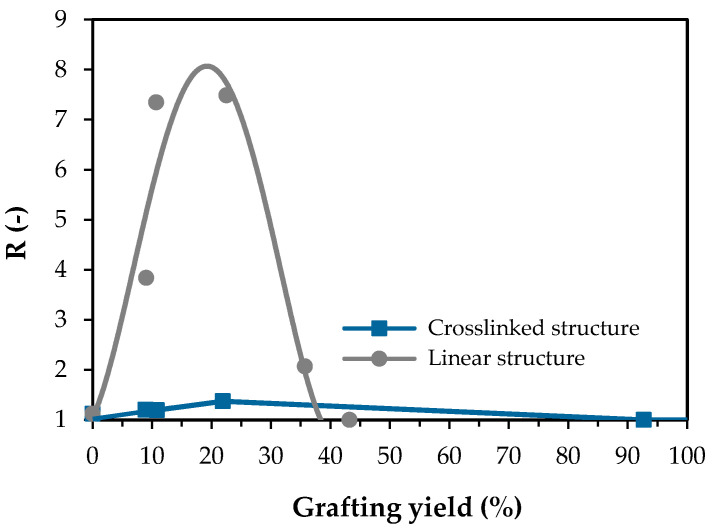
Effect of grafting yield on the thermoresponsive gating characteristics of PNIPAm-grafted nylon membranes of both architectures.

**Figure 9 polymers-15-00497-f009:**
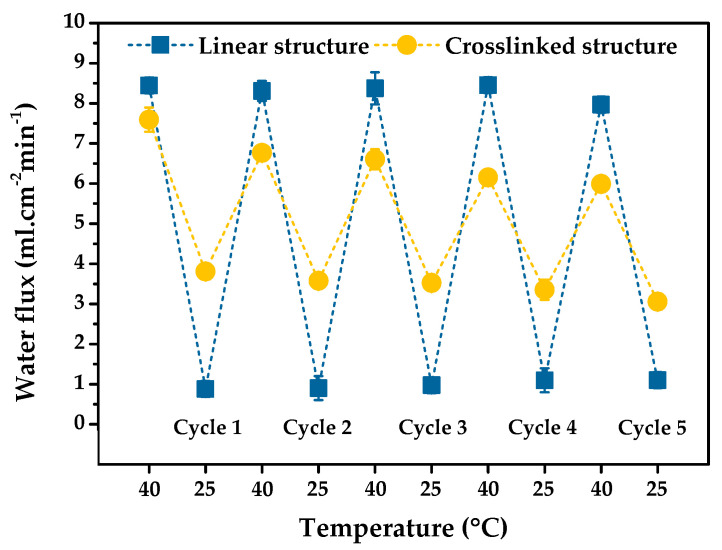
Reversibility of the thermoresponsive “open-closed” switching function of the grafted PNIPAm gates in the membrane pores under the test pressure of 100 kPa. (The optimum grafting yield of 22.5 and 21.9% is for the linear and crosslinked PNIPAm structures).

**Figure 10 polymers-15-00497-f010:**
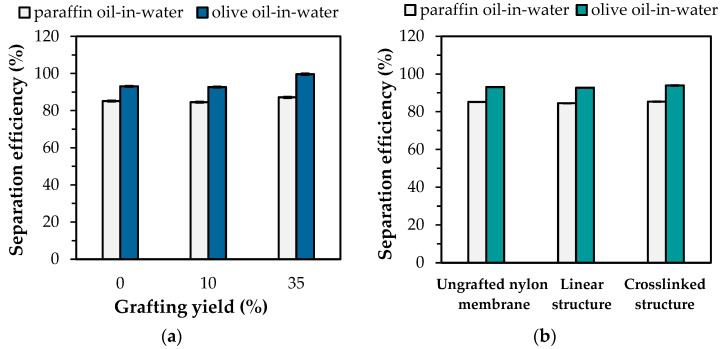
The separation efficiencies of O/W emulsions at 25 °C of: (**a**) linear PNIPAm-grafted nylon membrane with different grafting yields; (**b**) PNIPAm-grafted nylon membrane with different architectures.

**Figure 11 polymers-15-00497-f011:**
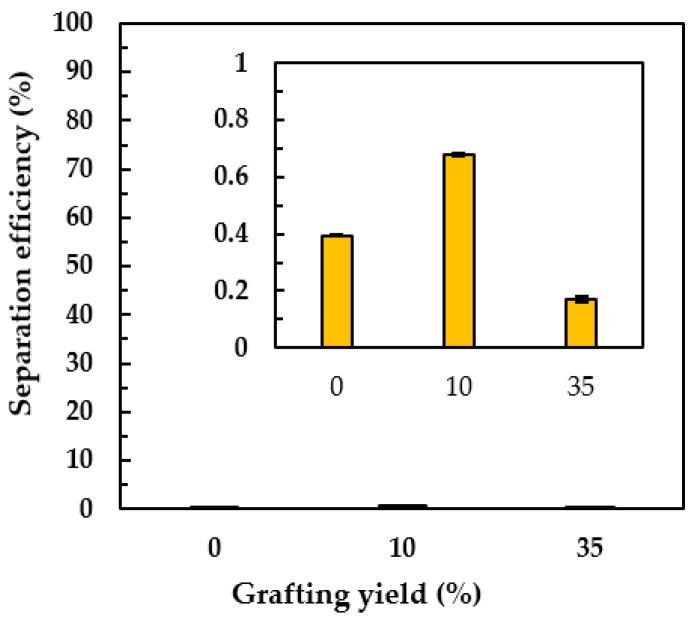
The separation efficiencies of the oil-water mixture at 45 °C of linear PNIPAm-grafted nylon membrane with different grafting yields. The inset was the enlargement of the same graph to demonstrate the small but different separation efficiency of the membranes with different grafting yield.

**Figure 12 polymers-15-00497-f012:**
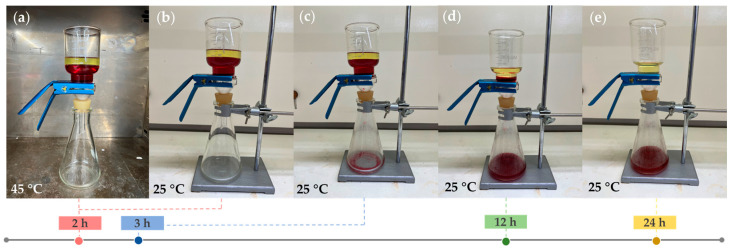
The images of oil-water mixture separation of linear PNIPAm-grafted nylon membrane with grafting yield of 35%; (**a**) the oil and water phases were obstructed above the linear PNIPAm-grafted nylon membrane and (**b**–**e**) the water separation process at below LCST of PNIPAm.

**Table 1 polymers-15-00497-t001:** Grafting yields of PNIPAm-grafted nylon membranes.

Grafting Yield (%)	
**Linear**	**Crosslinked**
6.2	7.3
9.0	8.9
10.7	10.8
17.3	15.5
22.5	21.9
35.7	92.7
43.2	154.4

**Table 2 polymers-15-00497-t002:** BET-specific surface area and total pore volume of linear PNIPAm-grafted nylon membranes.

Grafting Yield (%)	BET-Specific Surface (m^2^/g)	Total Pore Volume (cm^3^/g)
0	12.25	0.035
1–4	10.29	0.030
6–17	9.61	0.026
18–22	4.69	0.017

**Table 3 polymers-15-00497-t003:** Gas transmission rates and permselectivity of PNIPAm-grafted nylon membranes.

Structure	Grafting Yield (%)	O_2_TR (cm^3^/h)		CO_2_TR (cm^3^/h)		β	
		25 °C	35 °C	25 °C	35 °C	25 °C	35 °C
Ungrafted nylon	-	228 ± 26	258 ± 16	192 ± 14	223 ± 17	0.84 ± 0.04	0.86 ± 0.01
Linear	10.7	253 ± 14	329 ± 22	218 ± 7	276 ± 8	0.86 ± 0.02	0.84 ± 0.03
	43.2	31 ± 2	50 ± 2	35 ± 2	51 ± 1	1.13 ± 0.01	1.02 ± 0.05
Crosslinked	10.8	170 ± 37	253 ± 9	163 ± 21	218 ± 2	0.96 ± 0.11	0.86 ± 0.03
	92.7	28 ± 4	40 ± 1	31 ± 1	38 ± 1	1.11 ± 0.18	0.95 ± 0.03

## Data Availability

The data presented in this study are available upon request from the corresponding author.

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
