# Peer review of "Potential Applications of Thermoresponsive Poly(N-Isoproplacrylamide)-Grafted Nylon Membranes: Effect of Grafting Yield and Architecture on Gating Performance"

_polymers, 2023, doi:10.3390/polym15030497_

Round 1
Reviewer 1 Report
(1) In figure 8, please give more data to show the reversibility, 5 cycles are not enough to confirm the reversibility.
(2) From some research(APCBEE Procedia 8 (2014) 230-235), the abbreviation of oxygen, carbon dioxide, and water vapor transmission rates are OTR, CTR, and WVTR. In this paper, the oxygen and carbon dioxide transmission rates (OTR and CO2TR), please use the general abbreviation.
(3)The scale bar in Figure 3 is not clear, please show the clear scale bar in SEM images.
(4)Waht is the thermoresponsive mechanism of poly(N-isopropy- 2 lacrylamide), please illustrate the chemical structure change.
Author Response
Thank you for giving us the opportunity to submit a revised manuscript entitled “Potential Applications of Thermoresponsive poly(N-isopropylacrylamide)-grafted Nylon Membranes: Effect of Grafting Yield and Architecture on Gating Performance” for publication in Polymers journal. We appreciate the time and effort that you and the reviewers dedicated in providing valuable feedbacks on our manuscript. We have incorporated most of the suggestions made by the reviewers. Those changes were highlighted within the manuscript.
Please refer to the attached file for the responses to the reviewer's suggestions.

Reviewer 2 Report
Referee’s comments
To the paper entitled “Potential Applications of Thermoresponsive poly(N-isopropylacrylamide)-grafted Nylon Membranes: Effect of Grafting Yield and Architecture on Gating Performance” by Todsapol Kajornprai
The paper is devoted to very interesting problem, which is devoted to modifying polymer membrane to improve its separation ability. As for my opinion, the paper should be published after some additions.
First of all, I would propose to make the title more informative. “Effect of Grafting Yield and Architecture on gas separation” or something like this.
Please give detailed information about separation in the abstract. For instance, “The gas transmission rates through the PNIPAm-grafted nylon membranes increased…”. What gas? Some quantitative data are very welcome.
Grafting is a common approach to control hydrophilic-hydrophobic properties of membranes. Please mention also that the methods of their control for other membrane processes. For instance, baromembrane processes require hydrophilic membranes, hydrophilization is achieved by grafting hydrophilic agents (https://doi.org/10.1016/j.apsusc.2020.148905, https://doi.org/10.1016/j.msec.2021.112517) or insertion of inorganic ion-exchangers into pores of polymers (https://doi.org/10.2298/APT1647153M, https://doi.org/10.1016/j.memsci.2022.120984).
Additionally to the table, please also add pore size distributions and nitrogen adsorption isotherms.
Photos of liquid drop at the membrane surface are very desirable.
The SEM image of higher resolution is very welcome.
Author Response

(The authors gave the same response as above.)

Reviewer 3 Report
1. The similarity index is high (32 %) and should be reduced
2. In 2.2. part, the authors didn't refer to the preparation of crossslinked PNIP grafted nylon membrane
3. In part 2.3.5, do you mean the diameter ..was 11.34 or area?
4. In table 1, write the definite conc. corresponding to each grafting yield
5. in the experimental part, refer to the equation used for flux calculation as mentioned in:
6. https://onlinelibrary.wiley.com/doi/full/10.1002/vnl.21866
7. kindly, add a physical image for the oil/water separation system and examples of the contact angles measured
8. The change of contact angle of almost all samples, except at y= 16 % was under 90 o which is
9. still consider as hydrophilic CA. So, is it accurate to say (wettability shifted from hydrophilic to hydrophobic) or the hydrophilicity decreased.
10. Give a reference for your postulation (The presence of a crosslinker... polar groups)
11. You should apply statistical methods such as T-test to judge which Y % has a significant impact on CA because the change is so small in some cases.
12. Give a reference which support your explanation of the different behavior of linear and crosslinked PNIOAm- grafted nylon membrane represented in Fig. 6.

Author Response
Thank you for giving us the opportunity to submit a revised manuscript entitled “Potential Applications of Thermoresponsive poly(N-isopropylacrylamide)-grafted Nylon Membranes: Effect of Grafting Yield and Architecture on Gating Performance” for publication in Polymers journal. We appreciate the time and effort that you and the reviewers dedicated in providing valuable feedbacks on our manuscript. We have incorporated most of the suggestions made by the reviewers. Those changes were highlighted within the manuscript
Please refer to the attached file for the responses to the reviewer's suggestions.

Round 2
Reviewer 3 Report
No comments